# Critical Thinking in Initial Teacher Training: An Empirical Study from Chile

**DOI:** 10.3390/bs15050603

**Published:** 2025-05-01

**Authors:** Liliana Pedraja-Rejas, Christopher Maulén, Christofer Rivas

**Affiliations:** Departamento de Ingeniería Industrial y de Sistemas, Universidad de Tarapacá, Casilla 7D, Arica 1020000, Chile; cmaulenb@outlook.com (C.M.); crivasbi@outlook.com (C.R.)

**Keywords:** critical thinking, academics, diversity, achievement, higher education, universities

## Abstract

The promotion of critical thinking has become a key objective for educational institutions worldwide. In this context, it is essential to continue researching and promoting pedagogical strategies that favor its development. This study aims to explore academics’ perceptions of the characteristics of their academic units that could facilitate the development of critical thinking, as well as students’ perceptions of their own critical thinking skills, their levels of satisfaction, and the learning outcomes achieved. To achieve this objective, questionnaires were administered to 31 academics and 150 students at a Chilean university. The results revealed that, although academics assigned high scores to the three dimensions evaluated (gender diversity, academic preparation, and value congruence), when analyzing the data disaggregated by gender and school, significant differences in their perceptions emerged. On the other hand, in the case of students, the items related to facilities and equipment (satisfaction dimension) obtained the lowest scores. Likewise, no direct and uniform relationship was identified between the characteristics of academics and the development of critical thinking when grouping the data by school. Finally, a strong and significant correlation was observed between satisfaction, learning outcomes, and students’ perceived critical thinking skills. Several recommendations are presented to address the identified issues.

## 1. Introduction

The development of critical thinkers has become an essential goal for educational institutions worldwide ([46]). The ability to think critically not only fosters an open, problem-solving, and self-regulating mind, but also enables students to analyze information objectively, identify patterns, evaluate arguments, and make informed decisions. These skills not only have a direct impact on their academic performance, but also enhance their active participation in social and academic activities, strengthen their intellectual autonomy, and increase their opportunities for job placement and professional success ([26]; [52]).

In an increasingly globalized and complex world, where competitiveness and technological advances dictate social and economic dynamics, critical thinking is positioned as a key tool to face the challenges of the 21st century ([42]). Professionals with this skill are not only better prepared to solve problems and adapt to changes in the labor market but are also capable of innovating, proposing creative solutions, and leading transformation processes in their respective areas. Moreover, critical thinking is indispensable to form responsible and committed citizens, capable of actively contributing to the sustainable development of their communities and society as a whole ([4]; [12]; [21]).

Critical thinking is not only a requirement for personal or professional success; it is an essential competence to promote social awareness, empathy, and responsible transformation in an interconnected world. Without this ability, societies run the risk of perpetuating prejudice, misinformation, and irrational decisions. Therefore, deepening the development and teaching of critical thinking not only benefits individuals but also strengthens the foundations for more informed, reflective, and active citizenry ([46]; [49]).

In this context, it is imperative to continue researching and promoting educational strategies that enhance the development of critical thinking, an essential competence to prepare students to face the challenges of a world in constant transformation and contribute to the construction of more equitable, innovative, and sustainable societies. Despite its importance, the current literature reveals a notable lack of empirical studies that examine, in an integrated manner, student and academic perceptions of key factors such as academic preparation, gender diversity, value congruence, student satisfaction, and learning outcomes and their relationships with critical thinking.

This study addresses this knowledge gap and provides evidence from the Chilean context, expanding our understanding of how these elements interact in the development of this fundamental competency. To this end, this paper aims to explore academics’ perceptions of the characteristics of their academic units that could act as facilitators of the development of critical thinking, as well as students’ perceptions of their own critical thinking skills, their levels of satisfaction, and the learning outcomes achieved.

More specifically, the specific objectives of this research are (1) to analyze academics’ perceptions of key aspects of their department, such as gender diversity, the level of academic preparation, and value congruence, identifying possible differences by gender and school; (2) to explore students’ perceptions of their critical thinking skills, their levels of satisfaction, and the learning outcomes achieved; (3) to describe how the characteristics of academics (gender diversity, academic preparation, and value congruence) and critical thinking vary when grouped by school; and (4) to examine the relationship between student satisfaction, learning outcomes, and the critical thinking dimension from the perspective of the students themselves.

Understanding differences in academics’ perceptions of the characteristics of their academic units can help to identify possible biases or areas of dissatisfaction, which would allow for the development of more equitable and inclusive initiatives.

In addition, given that academic units vary in their characteristics and organizational culture, it is essential to analyze how these differences manifest themselves according to the school to which academics and students belong. This analysis will allow for the identification of strengths and areas for improvement.

Finally, exploring students’ perceptions will provide valuable information on their levels of satisfaction, the challenges that they face, and the learning outcomes achieved. These findings will be key to implement pedagogical strategies that are more aligned with their needs and expectations, thus contributing to more comprehensive and effective education.

## 2. Background

### 2.1. Critical Thinking

Critical thinking is defined as good judgment, as opposed to illogical or irrational thinking, and its development involves various cognitive skills. These abilities include interpretation, which is the ability to understand the context of a situation; analysis, which consists of reaching a conclusion about an issue; evaluation, which involves forming one’s judgment based on assessing the credibility of a statement; inference, which is the ability to draw conclusions, conjectures, and hypotheses from data, evidence, and beliefs; explanation, which is the ability to present the results of one’s reasoning reflectively and coherently; and self-regulation, which allows one to apply critical thinking to oneself to improve one’s thinking process ([14]).

In addition to cognitive skills, dispositions toward critical thinking play an essential role in the development of sound and reflective reasoning. These dispositions, which include open-mindedness, inquisitiveness, flexibility, fairness, perseverance in the search for truth, and confidence in reasoned inquiry processes ([14]), not only complement critical thinking skills but also influence how people apply these skills in various situations. In this sense, dispositions can be understood as stable internal motivations manifested in a commitment to critical thinking; cognitive maturity, reflected in an awareness of biases, values, and intellectual standards; and innovativeness, i.e., curiosity about truth and openness to new ideas ([9]). The importance of these dispositions lies in the fact that it is not enough to possess critical thinking skills; it is essential to be willing to use them when the situation requires it. Otherwise, a person could not be considered a critical thinker in the full sense ([5]).

The further development of critical thinking benefits students by empowering them with problem-solving skills ([3]) and enhances their creative abilities ([2]), among others. Although critical thinking is an indisputable educational goal, there is concern that, in practice, its development is not adequately fostered in students ([36]).

### 2.2. Characteristics of Academics

Academics participate in an integral way in a unit or school, playing a crucial role in the challenges of initial teacher training and fostering the development of critical thinking in students. Moore, based on the teaching perspective, identifies seven categories in the concept of critical thinking: making judgments; having a skeptical and questioning view of reality; being original and generating knowledge; reading sensibly and carefully; being rational and having a structured method of reasoning; adopting an ethical and activist stance; and practicing self-reflection and self-awareness ([4]).

Various studies indicate that there is no significant gender preference among students but highlight the importance of gender diversity in higher education. Successful male academics are rated more highly in aspects such as class leadership and knowledge of current topics, while female academics excel in course preparation and organization ([20]). The gender diversity of academics fosters creativity, innovation, and problem-solving—essential skills for the development of critical thinking ([30]; [31]; [34]). More gender-diverse academic teams can enhance students’ critical thinking, especially in mixed-gender classrooms, where same-sex academics may influence student interest ([20]). Various studies indicate that gender also impacts competencies such as design, practice, and thinking skills development ([12]). Additionally, the relationship between academic gender and socioeconomic status affects students’ perceptions, with academics with a similar status to students being more highly valued ([20]).

Academics have a strong influence on students’ academic performance ([17]). Therefore, teaching quality is essential in developing critical thinking; a well-trained academic can significantly motivate this development in students ([8]). Teacher training should include their levels of training, experience, and adaptability to the generational context, such as the handling of new technologies and cognitive openness—factors that directly impact students’ critical thinking ([28]). Some models suggest the path that a student should follow to develop critical thinking from fundamental levels, such as observing, reading, and informing, to advanced levels, such as discerning, making judgments, and proposing solutions ([4]).

Finally, in the realm of teaching, the development of ethical education and moral values is of the utmost importance. Academics must be congruent with what they teach and pay special attention to their ethical and moral values, considering that solid ethical and moral qualities will have a profound impact on an academic’s teaching ([44]).

### 2.3. Learning Outcomes

The learning process begins with acquiring knowledge, which refers to the capacity to store the content seen in classes, lectures, and presentations as memories. This process occurs mainly through communication between two people ([23]). Evidence shows that critical thinking relates to acquiring knowledge, as it favors the reception of knowledge imparted by an academic by developing skills such as inference, interpretation, and conclusion ([15]).

Thus, once knowledge is acquired, it should be used for problem-solving by applying it to related problem situations ([38]). Developing critical thinking provides a series of tools and skills for problem-solving, demonstrating their clear relationship ([30]).

The ability to self-learn refers, finally, to the ability to study and learn autonomously to acquire knowledge or access a job. This concept is also related to self-regulation, an essential characteristic of critical thinking ([51]). Research conducted in China evaluated the self-learning ability of a group of nursing students using the concept of “sandwich teaching”, a teaching model that alternates theoretical learning with work practice. Their results indicated that the students’ academic performance increased considerably as classes became theoretical–practical, where students were predisposed to discuss the subject matter, and the academic acted as a guide and tutor ([6]).

### 2.4. Satisfaction

Students’ perceptions of their levels of satisfaction are one of the most relevant outcomes in evaluating the educational process and, therefore, are as important as the learning achieved ([43]). Student satisfaction is the outcome of a subjective evaluation of their educational experiences and constitutes a positive antecedent of their loyalty ([50]).

Several authors have identified various factors that influence student satisfaction, such as the quality of the teaching–learning process; aspects related to infrastructure, educational resources, and facilities; personnel; and administrative processes ([24]). Critical thinking is also a valued skill that influences students’ perception of satisfaction ([47]).

## 3. Methods

This section describes the methodological framework of the study.

### 3.1. Study Design

This study has a descriptive approach as it seeks to provide an accurate description of a person, event, or situation without intervening or manipulating variables ([37]). Likewise, it has a correlational approach as it seeks to quantify the intensity of the linear relationship between the dimensions analyzed. The study’s design may reveal trends and associations between variables, but it does not establish causal relationships ([45]). Finally, this work is cross-sectional as it captures perceptions at a given time through questionnaires ([33]).

### 3.2. Participants

This study was carried out at a state university in the metropolitan region of Chile. The questionnaire was applied to 31 academics (5 men and 26 women) of 4 schools linked to the area of Humanities and Social Sciences. Additionally, a questionnaire was applied to 150 university students belonging to different undergraduate pedagogical programs.

The convenience sampling method was used, considering that the survey’s completion depended only on the participants’ availability. These instruments were applied in 2023.

Before participating, all academics and students received, in written form, detailed information about the objectives of the study, the procedures involved, the confidentiality of their responses, and their right to withdraw at any time without consequences. Subsequently, each participant voluntarily signed an informed consent form.

To guarantee the privacy and anonymity of the data, the responses were treated confidentially and analyzed in aggregate form, without the possibility of individually identifying the participants. In addition, the study followed established ethical norms and was conducted with the appropriate institutional authorization.

### 3.3. Measures and Data Collection

The questionnaires were elaborated on based on the state of the art. The dimension of gender diversity was constructed following the design of [22] ([22]), considering the proportion of women in faculty and management positions. The academic preparation was based on the work of [29] ([29]), covering undergraduate and graduate university studies, experience, and innovation capacity. Value congruence was based on the study of [10] ([10]), which integrates shared values, beliefs, and goals. Students’ development of critical thinking was assessed according to the work of [14] ([14]); it focuses on skills to interpret, analyze, represent, evaluate, explain, and self-regulate. The educational outcomes component considered the contributions of [16] ([16]) and [13] ([13]), including learning outcomes and student satisfaction.

The instrument intended for academics consisted of 10 statements (Appendix A), evaluated using a Likert scale ranging from 1 (strongly disagree) to 7 (strongly agree). The survey aimed to measure academics’ perceptions of gender diversity, the level of academic preparation, and value congruence.

Similarly, students were asked to complete another questionnaire to measure their perceptions of their learning outcomes, satisfaction, and critical thinking skills. The questionnaire consisted of 16 statements (Appendix B) and was evaluated using a 7-point Likert scale.

The questionnaires were administered face-to-face to academics, while a questionnaire was distributed to students through Google Forms using a QR code.

### 3.4. Reliability and Validity Tests

Cronbach’s alpha coefficient was used to evaluate the internal consistency of the questionnaires. It provides a measure of the relationship and reliability between the different items that constitute each dimension. Table 1 and Table 2 summarize the reliability of the instruments applied to academics and students, respectively. All dimensions gave alphas above 0.6, meaning that both questionnaires were acceptable and reliable.

### 3.5. Data Analysis

Once the surveys were collected, the data were structured in an Excel matrix and subsequently analyzed using the R Studio (version 2023.12.1-402) programming environment. The statistical analysis was carried out in three stages.

Descriptive analysis: Means and standard deviations (SD) were calculated to summarize the responses and identify general trends.

Comparative analysis: The Mann–Whitney–Wilcoxon test, a non-parametric method suitable for comparing independent groups when the assumption of normality is not met, was used to examine differences in the perceptions of academics according to gender. The significance level was set at 0.05.

Correlation analysis: To examine the possible relationships between critical thinking, student satisfaction, and learning outcomes, a correlation analysis was performed using Pearson’s coefficient. This method made it possible to assess the strength and direction of the associations between these variables, providing a quantitative basis for an understanding of their interaction. The force of the correlations was interpreted according to the following thresholds: no correlation (value less than 0.10), weak correlation (between 0.10 and 0.29), moderate correlation (between 0.30 and 0.50), and strong correlation (between 0.50 and 1.00) ([19]). Finally, *p*-values were calculated to evaluate the significance of the correlation coefficients.

## 4. Results

The following are the results of the study.

### 4.1. Descriptive Analysis of the Answers Given by Academics

Data were first processed for the descriptive analysis, since six questions contained null values. The corresponding average replaced the values.

The average scores per dimension were 5.27 (SD = 1.63) for gender diversity, 5.76 (SD = 0.81) for academic preparation, and 6.32 (SD = 0.61) for value congruence. Thus, value congruence had the highest mean score, while gender diversity had the lowest mean score. Table 3 shows the mean scores categorized by dimension, questionnaire item, and gender.

In the gender diversity dimension, as shown in the table, there is a difference, although not significant, in perceptions between the sexes: men tend to overestimate female representation in academic positions and underestimate their presence in leadership positions compared to the perceptions of their female colleagues, where the trend is reversed.

Question 5 stands out regarding academic preparation, as female academics score significantly lower than male academics. Thus, in this dimension, female academics feel that their inventive teaching endeavors are not acknowledged as highly as those of their male counterparts.

Moreover, in the value congruence dimension, question 8 is noteworthy. It shows that female academics have a significantly more positive perception than male academics, indicating that women feel a stronger connection to their educational role.

Table 4 shows the average scores categorized by dimension, questionnaire item, and school.

Table 4 shows that gender diversity is an area with potential for improvement, particularly since not all schools have equally positive perceptions. Based on this, there is a need to implement specific measures to address these perceptions and improve gender representation.

Schools value academic preparation, but specific areas, such as innovative teaching activities (question 5), may need review and potential improvement, especially in the FCB.

Value congruence is an area that is generally valued, although there are variations that could be addressed at the school level. For instance, question 10, which dealt with shared goals, shows lower average values than other questions. As a result, this indicates issues in this area that require specific attention and improvement, especially in the FAE school, where the lowest average was achieved.

### 4.2. Descriptive Analysis of Students’ Answers

The respondents comprised 58% females, 39% males, 2% who identified as non-binary, and 1% who chose the “other” alternative. The study participants ranged in age from 18 to 36 years and ranged from their first to their sixth year of academic training. The distribution by year of study was as follows: 27 first-year students, 18 second-year students, 32 third-year students, 35 fourth-year students, 36 fifth-year students, and 2 sixth-year students.

The average scores by dimension were 5.35 for critical thinking (SD = 1.02), 5.40 for satisfaction (SD = 0.98), and 4.93 for learning outcomes (SD = 1.24). Thus, satisfaction had the highest mean score, while learning outcomes had the lowest mean score.

Table 5 shows the mean scores categorized by dimension and questionnaire item. This table indicates that, in the dimensions of critical thinking and learning outcomes, all questions obtained similar scores, with a difference of less than 0.6 points. Nevertheless, regarding the satisfaction dimension, question 16 registered the lowest score. This score suggests that students are unsatisfied with the university’s facilities and equipment.

### 4.3. Characteristics of Academics and Critical Thinking According to Schools

The average values of the four key dimensions (gender diversity, academic preparation, value congruence, and critical thinking) in the different schools (FFE, FHG, FCB, and FAE) are presented in Table 6. It is observed that the perception of gender diversity is higher in FAE (6.38) and lower in FFE (4.83). As for academic preparation, FAE has the highest rating (6.06) and FCB the lowest (5.50). The congruence of values is assigned the highest score in FHG (6.50) and the lowest in FAE (6.00). Finally, students’ perceptions of critical thinking are higher in FFE (5.66) and lower in FAE (5.04). These results suggest that there are differences between the schools in these dimensions, which could reflect variations in the experiences and perceptions of the academics or students in each academic unit.

Figure 1 displays a graph of the different average scores for each dimension and school.

### 4.4. Correlational Analysis

Table 7 shows that both satisfaction and learning outcomes correlate strongly and positively with critical thinking. This statistically significant relationship suggests a proportional association between students’ perceived critical thinking skills and their satisfaction and learning outcomes.

## 5. Discussion

It is essential to prepare teachers to face the challenges of the 21st century, where they can play a crucial role in educating students to achieve academic and professional success ([25]). Teacher training programs should integrate critical thinking in all areas and specializations, given that it facilitates knowledge generation, promotes innovation, and can significantly improve the ability to solve complex problems at the societal level ([35]).

Several factors influence students’ development of critical thinking, including the characteristics of academics, such as gender diversity, academic preparation, and value congruence ([43]). Considering these issues, it was decided to identify the opinions of academics themselves on these issues.

Differences were found in male and female academics’ perceptions of gender diversity. The existence of gender gaps in academia has been well reported, and these gaps widen even more in academic positions ([7]). Therefore, women, by directly experiencing a lack of representation, may have a more realistic perception of the situation. Regardless, they value representation in leadership positions, which is remarkable considering that the still-existing stereotypes hinder the promotion of women to these positions ([48]).

Regarding academic preparation, female academics felt that their innovative teaching efforts were not acknowledged as highly as those of their male peers. The literature highlights that women tend to be more represented in activities that, within academic reward systems, are considered less “valuable”, such as teaching ([41]). Their underrepresentation in more valued areas may explain why they perceive less recognition for their contributions. However, as a consequence of this low recognition, they expressed a strong commitment when asked about their commitment to teaching, indicating that they continue to value teaching as fundamental.

In assessing these dimensions by school, disparities in perceptions were evident. The issue of innovation in teaching (academic preparation) emerged as one of the areas with the lowest scores. It is imperative to review and potentially reform academic recognition criteria and processes to ensure that they are fair and transparent. Furthermore, differences were observed in the perceptions of academics regarding shared goals. Given that alignment between institutional goals and the goals of academics leads to desirable pedagogical outcomes, and that an academic with shared values facilitates the holistic development of students ([43]), it is recommended to focus on defining and communicating shared goals. Encouraging a sense of common purpose and collaboration among academic staff is crucial in fostering stronger cohesion and effectiveness in teaching.

Student perceptions generally scored highly in the dimensions of critical thinking, satisfaction, and learning outcomes. However, the satisfaction dimension contained the item with the lowest score. Infrastructure and facilities have been identified as key indicators of service quality in higher education ([1]). Furnishings also play a crucial role in the learning environment and learning experience and are linked to good health, improved academic performance, and motivation ([39]). Higher education institutions must ensure the availability of top-quality infrastructure and facilities, as this not only enables efficient learning and performance but also motivates their employees to deliver excellent services ([1]). Moreover, for students, it fosters a connection with their campus and enthusiasm to use the services, which can directly impact their satisfaction levels ([27]).

When analyzing academics’ perceptions of gender diversity, academic preparation, and value congruence, as well as student perceptions of critical thinking, the results showed significant differences between schools, suggesting that these groups’ experiences and perceptions vary according to the academic context. Although the literature has proposed a relationship between the characteristics of the academic body and the development of critical thinking ([43]), the findings in this sample do not show a clear trend in this regard.

For example, although FAE reported the highest perception of gender diversity, its students assigned relatively low scores to the development of critical thinking. In contrast, FFE scored the highest in critical thinking, but it exhibited one of the lowest perceptions of gender diversity. Similar situations were evident when analyzing other variables, such as academic preparation and value congruence. These results suggest that the relationship between the characteristics of academics and the development of critical thinking is not linear. Instead, it could be mediated by other factors, such as the organizational culture, teaching methodologies, student expectations, or even institutional dynamics, which were not directly measured in this study.

Finally, a significant positive correlation was identified between critical thinking and the dimensions of satisfaction and learning outcomes. From an applied perspective, the findings of this study underscore the importance of academics orienting their teaching toward developing critical thinking skills. Critical thinking skills are not only related to knowledge acquisition ([15]) but also to problem-solving ([18]; [30]) and self-learning skills ([6]). Critical thinking also contributes to higher levels of satisfaction among students ([32]). Indeed, this is crucial, as student satisfaction is an essential determinant in evaluating educational quality and can provide a competitive advantage to institutions ([11]; [40]).

## 6. Limitations

This study has three main limitations that may influence the interpretation and applicability of the results.

First, the small number of academics who responded to the survey may had limited the diversity of perspectives, as variations inherent to different academic profiles, such as levels of experience, areas of specialization, or career paths, were not sufficiently captured. This could have biased the findings toward opinions representative of a specific group, reducing the study’s ability to reflect the overall reality of academics.

Second, conducting the study at a single university restricted its geographic and institutional scope. This implies that the conclusions were influenced by the characteristics of this institution, such as its organizational culture, available resources, internal policies, and student composition. Without the inclusion of universities of different sizes, geographic locations, and socioeconomic contexts, the results lack the heterogeneity necessary to generate a more complete picture of educational dynamics.

Third, the cross-sectional design of the study, based on applied surveys, only allowed us to capture associations at a specific point in time, without providing information on how these relationships might develop or transform over time.

Finally, the descriptive nature of the study, while valuable in identifying initial trends and relationships, precluded the establishment of causal associations or robust statistical inferences. This limits the ability to apply the findings to other educational settings. Without a more advanced analytical approach, such as multivariate or longitudinal models, it is not possible to accurately determine whether the observed results are consistent or generalizable to different contexts.

These limitations highlight the need to interpret the results with caution and underscore the importance of future studies that address these issues to strengthen the external validity and practical impact of educational research.

## 7. Conclusions

Considering the importance of developing critical thinking skills, this study explored the perceptions of students and academics at a university regarding this skill and theoretically associated variables, such as the characteristics of academics, learning outcomes, and student satisfaction.

The results showed that, from academics’ perspective, value congruence is the most valued, while gender diversity is the least appreciated, according to the average scores of the responses. Notably, the average scores were high (above five on a scale of 7) in the three dimensions analyzed, providing reassurance about the overall performance. However, when the data were disaggregated according to gender and school, differences in perceptions were observed, indicating areas for potential improvement.

Although the students’ averages for the dimensions evaluated (critical thinking, satisfaction, and learning outcomes) were high, the items related to facilities and equipment obtained the lowest scores. Considering that both are critical components of the educational process and can affect students’ well-being, satisfaction, and performance, it is recommended that they be reviewed or repaired and investments made in improving existing equipment.

Likewise, it was identified that the relationship between the characteristics of academics and the development of critical thinking in students is neither direct nor homogeneous across schools. This suggests the possible influence of multiple contextual factors. Therefore, it is essential to adopt a comprehensive approach in teacher training and in the design of educational strategies, considering not only the individual characteristics of academics but also the academic context in which learning takes place.

Lastly, a strong and significant correlation was found between satisfaction and learning outcomes and students’ perceived critical thinking skills. These findings underscore the importance of further analysis to deepen our understanding of this relationship and its implications in the educational setting, emphasizing the need for continued research in this area.

## Figures and Tables

**Figure 1 behavsci-15-00603-f001:**
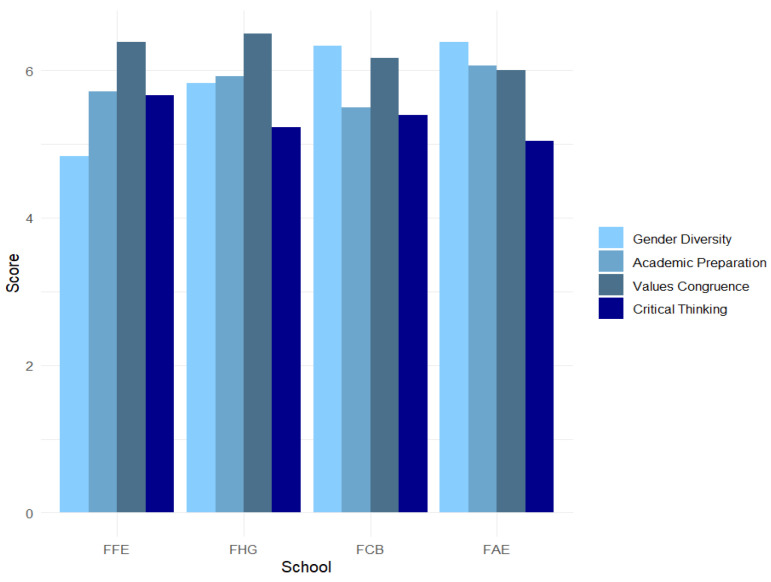
Mean perceptions of characteristics of academics and critical thinking by school.

**Table 1 behavsci-15-00603-t001:** Reliability of the questionnaire applied to academics.

Dimension	Cronbach’s Alpha
Gender diversity	0.72
Academic preparation	0.61
Value congruence	0.61

**Table 2 behavsci-15-00603-t002:** Reliability of the questionnaire applied to students.

Dimension	Cronbach’s Alpha
Critical thinking	0.87
Satisfaction	0.85
Learning outcomes	0.69

**Table 3 behavsci-15-00603-t003:** Average dimension scores by gender.

Dimension	Item	Gender				W	*p*-Value
		Male		Female			
		Mean	SD	Mean	SD		
Gender Diversity	1	5.4	2.07	4.92	2.17	69.5	0.821
	3	5	1.87	5.65	1.44	53.5	0.538
Academic Preparation	4	6	1.22	6.46	0.76	50.5	0.398
	5	6	1	3.97	1.82	110	0.014 *
	6	6	1.22	6.3	0.83	56	0.618
	7	5.8	1.09	6.08	0.86	84.5	0.288
Value Congruence	2	7	0	6.77	0.58	75	0.381
	8	5.8	0.84	6.28	0.93	104	0.030 *
	9	6	0.7	6.27	1.04	79	0.448
	10	5.4	0.89	6.12	1.1	40	0.154

* *p* < 0.05.

**Table 4 behavsci-15-00603-t004:** Average dimension scores by school.

Dimension	Item	School of Philosophy and Education (FFE)		School of History, Geography, and Letters (FHG)		School of Sciences (FCB)		School of Arts and Physical Education (FAE)		Total	
		Mean	SD	Mean	SD	Mean	SD	Mean	SD	Mean	SD
Gender Diversity	1	4.33	2.06	5.33	2.89	6.66	0.58	7.00	0.00	5.83	1.38
	3	5.33	1.62	6.33	0.58	6.00	1.73	5.75	1.26	5.85	1.30
Academic Preparation	4	6.24	0.94	6.67	0.57	6.67	0.57	6.75	0.50	6.58	0.65
	5	4.14	2.03	4.67	1.15	3.67	2.08	5.67	1.15	4.54	1.60
	6	6.28	0.95	6.67	0.57	6.00	1.00	6.00	0.81	6.24	0.83
	7	6.20	0.76	5.67	0.57	5.67	1.53	6.25	0.95	5.95	0.95
Value Congruence	2	6.76	0.62	6.67	0.58	7.00	0.00	7.00	0.00	6.86	0.30
	8	6.28	0.90	6.67	0.58	6.00	1.41	6.00	0.82	6.24	0.93
	9	6.28	1.06	6.67	0.58	5.33	0.58	6.25	0.96	6.13	0.79
	10	6.20	0.98	6.00	1.00	6.33	0.58	4.75	1.50	5.82	1.01

**Table 5 behavsci-15-00603-t005:** Average dimension scores.

Dimension	Item	Mean	SD
Critical Thinking	2	5.23	1.34
	3	5.39	1.31
	5	5.67	1.16
	7	5.25	1.36
	11	5.41	1.37
	15	5.15	1.33
Satisfaction	1	5.29	1.39
	4	5.65	1.13
	6	5.95	1.54
	8	5.00	1.35
	10	5.91	1.35
	12	5.79	1.13
	14	5.39	1.67
	16	4.27	1.57
Learning Outcomes	9	4.73	1.46
	13	5.14	1.38

**Table 6 behavsci-15-00603-t006:** Mean perceptions of gender diversity, academic preparation, value congruence, and critical thinking by school.

School	Gender Diversity	Academic Preparation	Value Congruence	Critical Thinking
FFE	4.83	5.71	6.38	5.66
FHG	5.83	5.92	6.50	5.23
FCB	6.33	5.50	6.17	5.39
FAE	6.38	6.06	6.00	5.04

**Table 7 behavsci-15-00603-t007:** Correlation matrix between critical thinking, student satisfaction, and learning outcomes.

	Critical Thinking	Satisfaction	Learning Outcomes
Critical Thinking	-		
Satisfaction	0.75 ***	-	
Learning Outcomes	0.74 ***	0.57 ***	-

*** *p* < 0.001.

## Data Availability

Data available on request from the corresponding author due to privacy and ethical reasons.

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
