# Peer review of "Critical Thinking in Initial Teacher Training: An Empirical Study from Chile"

_behavsci, 2025, doi:10.3390/bs15050603_

Round 1

Reviewer 1 Report

Comments and Suggestions for Authors
  • The manuscript is scientifically sound and the experimental design is adequate to test the hypothesis.
  • The results of the manuscript are reproducible based on the details provided in the methods section.
  • The figures/tables/images/schemes are appropriate. They present the data correctly. They are easy to interpret and understand. The data is interpreted appropriately and consistently throughout the manuscript. Include details about the statistical analysis or the data acquired from specific databases.
  • The conclusions are consistent with the evidence and arguments presented.
  • Evaluate the ethics statements.
  • General questions to help guide your review report for review articles:
  • The review is clear, comprehensive and relevant to the field. A gap in knowledge is identified.
  • What needs to be improved:
  • The authors do not mention the importance of critical thinking dispositions (lines 62 to 76).
  • Participants: They should mention the number of students per year of study (lines 254-255) and analyse whether there are statistically significant differences in means.
  • The questionnaire applied to students should be subjected to a factorial analysis in order to strengthen the analysis. The same cannot be done for the teachers' questionnaire due to the small number of responses.
  • Line 171, reference Hierro (2029) is missing

Author Response

Reviewer 1

Comment 1: The manuscript is scientifically sound and the experimental design is adequate to test the hypothesis.

The results of the manuscript are reproducible based on the details provided in the methods section.

The figures/tables/images/schemes are appropriate. They present the data correctly. They are easy to interpret and understand. The data is interpreted appropriately and consistently throughout the manuscript. Include details about the statistical analysis or the data acquired from specific databases.

The conclusions are consistent with the evidence and arguments presented.

Evaluate the ethics statements.

Response 1: We appreciate your positive comments and your detailed evaluation of our manuscript. We are pleased to know that you consider the experimental design to be adequate, the results to be reproducible, and the figures/tables to be clear and appropriate.

In response to your comments, we have included additional details on the statistical analysis in section 3.5, specifying the tests used and the criteria applied for data interpretation. We have also updated the data availability statement in the corresponding section, as requested by the editor. It is worth mentioning that in this study we used primary data, which were collected and processed following the protocols described in the manuscript.

Regarding the ethical statements, we confirm that the study was conducted in compliance with all required ethical standards. Prior to the start of the study, formal approval was obtained from the corresponding ethics committee. In addition, we ensured that all participants signed their written informed consent, which detailed the objectives of the study, the procedures, the confidentiality of the data, and the right to withdraw at any time without consequences. These statements are detailed at the end of the manuscript.

Comment 2: General questions to help guide your review report for review articles:

The review is clear, comprehensive and relevant to the field. A gap in knowledge is identified.

Response 2: We appreciate and thank you for your kind words.

Comment 3: What needs to be improved: The authors do not mention the importance of critical thinking dispositions (lines 62 to 76).

Response 3: Dear reviewer, thank you for your comment. We have incorporated a new paragraph in section 2.1 to highlight the importance of dispositions toward critical thinking. In this revision, key references from Facione (2020), Butler (2024) and Chen et al. (2024) have been integrated to strengthen the argument.

Comment 4: Participants: They should mention the number of students per year of study (lines 254-255) and analyse whether there are statistically significant differences in means.

Response 4: Dear reviewer, thank you for your comment. We have included in the text the detailed distribution of the number of students by year of study, which is as follows: 27 first year students, 18 second year students, 32 third year students, 35 fourth year students, 36 fifth year students and 2 sixth year students.

Regarding the analysis of statistically significant differences in the means, we consider that it is not appropriate to perform this analysis due to the imbalance in the size of the groups and the reduced number of participants in some years (especially in the sixth year, with only 2 students).

Comment 5: The questionnaire applied to students should be subjected to a factorial analysis in order to strengthen the analysis. The same cannot be done for the teachers' questionnaire due to the small number of responses.

Response 5: Dear reviewer, we appreciate your observation and suggestion regarding the performance of a factor analysis for the questionnaire applied to the students. We understand that this analysis could provide greater robustness to the study by exploring the underlying structure of the constructs assessed. However, in this case, we have chosen not to perform a factor analysis due to the following reasons:

- Although we had 150 students, we consider that this number might be insufficient for a reliable factor analysis, as a larger sample size is recommended to ensure the stability of the results.

- Our main objective was to analyze student perceptions and explore possible correlations between the variables (critical thinking skills, satisfaction and learning outcomes), rather than to validate the factor structure of the questionnaire.

- To evaluate the reliability of the instrument, we relied on Cronbach's alpha, which showed acceptable values for each of the scales, indicating adequate internal consistency.

Regarding the teachers' questionnaire, we agree that the small number of responses does not allow for a reliable factor analysis, so we have presented the results in a descriptive and exploratory manner.

We recognize that a factor analysis could be a methodological improvement for future studies, and we thank you for your suggestion, which we will take into account for further research.

Comment 6: Line 171, reference Hierro (2029) is missing

Response 6: Dear reviewer, we have reviewed the indicated citation and we confirm that it is properly referenced. The author's correct last name is del Hierro and the reference corresponds to the year 2019.

Reviewer 2

I have several comments that might be addressed:

Comment 1: The objectives or hypothesis or questions should be moved to the end of the Introduction section and be removed from the methodology section.

Response 1: We appreciate the reviewer's comment. Following his suggestion, we have moved the objectives to the end of the introduction section and removed them from the methodology section.

Reviewer 2 Report

Comments and Suggestions for Authors

I have several comments that might be addressed:

1) The objectives or hypothesis or questions should be moved to the end of the Introduction section and be removed from the methodology section.

2) The objective 1 should be reformulated since as it is, does not refer to the critical thinking development. Note that in the intro section a clear definition of teachers in terms of critical thinkers is formulated. Therefore you should be consistent. In addition in the abstract it seems that the students perceptions on teachers instruction and academics perceptions are too separated. You should provide clear links between them.

3) Please provide some sentences related to the consentment of participants in participating in the study, provide some ethics consideration and how data was treated (for both teachers and students).

4) I consider that the categories for academics: gender diversity, academic preparation, and values congruence are disconencted to the students skills. You should link both studies for readers to connect the academics' competences to the students' ones on critical thinking. As it is, this study seems the sum of two experiments with low linkage. For example, how the academic preparation is related to students' critical thinking. I cannot cope with the instruction, or the methodologies the academics use to promote students' critical thinking development.

5) Then in table 4, you initiate an interesting faculty study, which is not considered in the Abstract or in the Introduction sections.

6) In the same line, you should provide the links between academic procedures/techniques with the dimensions for students development. I quite do not understand, the categories. I mean, critical thinking for what, satisfaction in relation to self-development, teachers instruction, etc.

7) The Discussion section should stress the academics/students processes in relation to the development of critical thinking, which seems not to be fully developed in the present narrative.

Author Response

Reviewer 2

I have several comments that might be addressed:

Comment 1: The objectives or hypothesis or questions should be moved to the end of the Introduction section and be removed from the methodology section.

Response 1: We appreciate the reviewer's comment. Following his suggestion, we have moved the objectives to the end of the introduction section and removed them from the methodology section.

Comment 2: The objective 1 should be reformulated since as it is, does not refer to the critical thinking development. Note that in the intro section a clear definition of teachers in terms of critical thinkers is formulated. Therefore you should be consistent. In addition in the abstract it seems that the students perceptions on teachers instruction and academics perceptions are too separated. You should provide clear links between them.

Response 2: We are grateful for the comments received. After analysing them, we have decided to maintain objective 1, considering it fundamental to the study, but we have enriched it with more detail. In particular, we have incorporated an analysis that assesses differences according to the gender of the academic and the school to which he/she belongs.

Also, in response to the reviewer's comment, we have added a fourth objective that seeks to examine how academics‘ perceptions of their characteristics and critical thinking vary from the students’ perspective, comparing results across schools. Given the small number of teachers (n=4), this analysis is approached descriptively, as the sample imbalance precludes statistically significant correlations or comparisons of means.

Finally, the abstract has been rewritten to avoid excessive separation between students' perceptions of teacher instruction and their academic perceptions, thus achieving greater coherence and connection between the two aspects.

Comment 3: Please provide some sentences related to the consentment of participants in participating in the study, provide some ethics consideration and how data was treated (for both teachers and students).

Response 3: We appreciate your comment regarding the inclusion of ethical considerations and the consent of the participants in our study.

In response to your suggestion, we have incorporated details of the procedure followed to obtain informed consent from teachers and students at the end of section 3.2.

We have also added information on the treatment of the data, emphasizing that they were handled confidentially and analyzed in aggregate form to guarantee the anonymity of the participants.

Comment 4: I consider that the categories for academics: gender diversity, academic preparation, and values congruence are disconected to the students skills. You should link both studies for readers to connect the academics' competences to the students' ones on critical thinking. As it is, this study seems the sum of two experiments with low linkage. For example, how the academic preparation is related to students' critical thinking. I cannot cope with the instruction, or the methodologies the academics use to promote students' critical thinking development.

Response 4: We appreciate the reviewer's comment and, as mentioned in comment 2, we incorporate a new objective that seeks to link the characteristics of the academics with students' critical thinking. While we recognise the relevance of this topic, the aim of this study is not to analyse in depth how the instruction or methodologies used by academics promote the development of critical thinking in students, but to examine differences in the perception of characteristics of the academics, understood as gender, academic preparation and values congruence. From a theoretical approach, these aspects could influence students' critical thinking, which justifies their inclusion in the analysis. Although our study is mainly descriptive, it provides an empirical basis that allows us to open the discussion on this topic and its relevance for future research. Furthermore, in the discussion section we have improved the integration of these results.

Comment 5: Then in table 4, you initiate an interesting faculty study, which is not considered in the Abstract or in the Introduction sections.

Response 5: Dear reviewer, we deeply appreciate your comment and accept your suggestion. As mentioned above, the main objective of the study was reworded to make it clear that the perceptions of academics are compared according to gender and school. Also, new paragraphs were added at the end of the introduction to emphasize the relevance of the study and to provide a broader context.

Among these additions, we highlight the importance of conducting a comparative analysis by faculty, as this approach allows us to identify both strengths and specific areas for improvement in each academic unit. We value your feedback, as it has allowed us to enrich the clarity and coherence of the manuscript.

Comment 6: In the same line, you should provide the links between academic procedures/techniques with the dimensions for students development. I quite do not understand, the categories. I mean, critical thinking for what, satisfaction in relation to self-development, teachers instruction, etc.

Response 6: We appreciate the reviewer's comment. As mentioned in previous responses, clearer links are now established between the characteristics of the academic body and the dimension of students' critical thinking, approaching this relationship from a descriptive perspective. In addition, the methodology section (item 5.3) and Appendix 1 specify which aspects were considered to measure each dimension, providing greater clarity on their operationalization.

Comment 7: The Discussion section should stress the academics/students processes in relation to the development of critical thinking, which seems not to be fully developed in the present narrative.

Response 7: In response to the reviewer's comment, two new paragraphs have been added to the Discussion section. These paragraphs discuss the results obtained by comparing the characteristics of academics and the level of critical thinking by school. Also, to strengthen the coherence of the manuscript, a new paragraph has been incorporated in the conclusion that synthesizes these findings and their relevance for educational practice. Finally, the abstract has been rewritten to more accurately reflect the contributions of the study.

Round 2

Reviewer 2 Report

Comments and Suggestions for Authors

No more suggestions for authors.